# Oxytocin Receptors on Calvarial Periosteal Innervation: Therapeutic Target for Post-Traumatic Headache?

**DOI:** 10.3390/pharmaceutics16060760

**Published:** 2024-06-04

**Authors:** Vimala N. Bharadwaj, Michael Klukinov, Robert Paul Cowan, Nazanin Mahinparvar, David John Clark, David Clifford Yeomans

**Affiliations:** 1Department of Anesthesiology, Perioperative and Pain Medicine, School of Medicine, Stanford University, Stanford, CA 94305, USAklukinov@stanford.edu (M.K.); robertpcowan@gmail.com (R.P.C.); mahinparvar@stanford.edu (N.M.); djclark@stanford.edu (D.J.C.); 2Anesthesiology Service, Veterans Affairs Palo Alto Health Care System, 3801 Miranda Ave (E4-220), Palo Alto, CA 94304, USA

**Keywords:** post-traumatic headache (PTH), mild traumatic brain injury, oxytocin, oxytocin receptor, trigeminal ganglia, periosteum, periosteal trigeminal afferent

## Abstract

Objective: Following a mild traumatic brain injury (mTBI), the most prevalent and profoundly debilitating occurrence is the emergence of an acute and persistent post-traumatic headache (PTH), for which there are presently no approved treatments. A crucial gap in knowledge exists regarding the consequences of an mTBI, which could serve as a foundation for the development of therapeutic approaches. The activation of trigeminal sensory nerve terminals that innervate the calvarial periosteum (CP)—a densely innervated tissue layer covering the calvarial skull—has been implicated in both migraines and PTHs. We have previously shown that trigeminal oxytocin receptors (OTRs) may provide a therapeutic target for PTHs. This study examined the expression of oxytocin receptors on trigeminal nerves innervating the periosteum and whether these receptors might serve as a therapeutic target for PTHs using a direct application of oxytocin to the periosteum in a rodent model of PTH. Methods: We used retrograde tracing and immunohistochemistry to determine if trigeminal ganglion (TG) neurons innervating the periosteum expressed OTRs and/or CGRPs. To model the impact of local inflammation that occurs following an mTBI, we applied chemical inflammatory mediators directly to the CP and assessed for changes in immediate-early gene expression as an indication of neuronal activation. We also determined whether mTBI would lead to expression changes to OTR levels. To determine whether these OTRs could be a viable therapeutic target, we assessed the impact of oxytocin injections into the CP in a mouse model of PTH-induced periorbital allodynia. Results: The results of these experiments demonstrate the following: (1) the cell bodies of CP afferents reside in the TG and express both OTRs and CGRPs; (2) inflammatory chemical stimulation of the periosteum leads to rapid activation of TG neurons (phospho-ERK (p-ERK) expression), (3) mTBI-induced inflammation increased OTR expression compared to the sham group; and (4) administration of oxytocin into the periosteum on day 2 and day 40 blocked cutaneous allodynia for up to one hour post-administration for both acute and persistence phases in the PTH model—an effect that was preventable by the administration of an OTR antagonist. Conclusion: Taken together, our observations suggest that periosteal trigeminal afferents contribute to post-TBI craniofacial pain, and that periosteum tissue can be used as a potential local target for therapeutics such as oxytocin.

## 1. Introduction

A post-traumatic headache (PTH) often ensues as a result of mild traumatic brain injury (mTBI) [1], with about 90% of all mTBI patients developing PTHs [2]. For some patients, these headaches resolve in the first 3 months after an injury and are referred to as acute PTHs. However, for about 40% of patients with acute PTHs, the headaches persist for several months, a condition termed persistent PTH (PPTH) [2,3]. At present, there have been no clinical trials or FDA-approved drugs specific to PTHs or PPTHs [1,4]. Despite having similar symptoms, current treatments developed for migraines or tension-type headaches [5] have limited efficacy and possess undesirable side effects [6,7,8,9].

One of the major obstacles to developing effective PTH therapeutics is a critical knowledge gap in understanding the sequelae to mTBI. An improved understanding of this pathophysiology can lead to the discernment of novel targets for developing PTH therapeutic strategies. However, it is clear that the activation of the trigeminal nerve system contributes to PTHs. Branches of the trigeminal nerve innervate the calvarial periosteum (referred to as periosteum henceforth), a dense layer of vascular and innervated connective tissue overlying the calvarial skull [10,11,12]. The contribution to PTHs of these afferents, which are, by definition, more exposed to skull impact is unclear. However, this exposure may enable an extracephalic therapeutic approach wherein treatments are delivered directly to trigeminal endings by periosteal administration in order to attenuate the activation of these afferents. This contention is supported by the results of previous studies which demonstrated that trigeminal nerve fibers innervating the periosteum are composed mainly of pain-sensing nerve fibers [10,13]. Specifically, unmyelinated fibers with calcitonin gene-related peptide (CGRP)—a marker for sensory pain nerve fibers—have been reported in the periosteum [10]. Preclinical and clinical studies provide extensive evidence that the CGRP plays a critical role in the pathophysiology of migraines [14,15] and has also been associated with PTHs [16,17]. The activation of the periosteal trigeminal sensory nerve terminals has been suggested to play a role in migraines and PTHs [13,18,19]. For example, the activation of periosteal nociceptive afferents produces PTH-like allodynia and promotes persistent inflammation in the periosteum, sensitizing the afferents [20]. Furthermore, human studies corroborate the findings in rodents, revealing that biopsied periosteum tissue samples from patients with chronic migraines show upregulation of pro-inflammatory genes and downregulation of anti-inflammatory genes [18]. It is not known however whether these phenomena are also present following an mTBI-induced PTH. Taken together, these studies provide presumptive evidence that targeting the periosteum as a drug delivery site may be promising for treating P/PTH.

Research conducted by our laboratory and other research groups has shown that the interaction between oxytocin and oxytocin receptors (OTRs) on trigeminal ganglion (TG) neurons in vitro resulted in a reduction in their excitability [21,22]. In particular, we have found that oxytocin increased the resting membrane potential and rheobase of trigeminal neurons taken from rats with a pre-existing craniofacial injury, both of which indicate a clear decrease in excitability [21], likely mediated by an increase in the potassium current [22]. In addition, the application of nasal oxytocin reduced PTH-related pain in mTBI-induced rats [23]. We hypothesize periosteal afferents, like most other trigeminal nociceptors, should possess OTRs and that direct calvarial periosteum delivery of oxytocin will significantly reduce the primary afferent activity and sensitization, and thus potentially reduce mTBI-related craniofacial pain. The rationale for this hypothesis lies in the known effects of oxytocin on nociception and inflammation. Oxytocin has been shown to modulate pain by interacting with OTRs on sensory neurons, leading to a reduction in pain signaling. Additionally, oxytocin has anti-inflammatory properties, which could help to lower the inflammatory response following an mTBI, a common contributor to craniofacial pain. By reducing primary afferent activity and sensitization, oxytocin delivery to the calvarial periosteum might provide a targeted therapeutic approach for managing craniofacial pain after an mTBI. This mechanism could help to minimize the intensity of post-traumatic headaches and related discomfort, offering a promising intervention for those suffering from these conditions.

The objective of this study is to determine whether calvarial periosteal neurons express OTRs and if they are a useful target for the inhibition of PTH-related pain. The study’s specific aims are to determine whether: (1) periosteal stimulation activates trigeminal ganglion (TG) neurons, (2) trigeminal neurons and fibers in the periosteum express oxytocin receptors (OTRs) and/or calcitonin gene-related peptides (CGRPs), (3) inflammation from mild traumatic brain injury (mTBI) increases OTR expression, and (4) post-traumatic headache (PTH)-related pain can be reduced by injecting oxytocin into periosteal tissue after an mTBI, with OTR antagonists negating this analgesic effect.

## 2. Materials and Methods

Materials: Oxytocin (Sigma, St. Louis, MO, USA catalog # O4375), OTR antagonist, L-368,899 (# L2540, Sigma), Atosiban (Millipore Sigma catalog # A3480), phosphor ERK antibody (#MAB1576; R&D Systems, Invitrogen, Waltham, MA, USA), OTR antibody (# ab217212, Abcam, Cambridge, UK), CGRP antibody (ab 36001, Abcam), secondary antibody, anti-mouse 546 (# A10036), anti-rabbit 488 (#A21206, Invitrogen), anti-goat 647 (#A21447, Invitrogen), inflammatory mediator soup: Bradykinin (1 mM), histamine (1 mM), serotonin (1 mM), prostaglandin E2 (100 μM) in SIF (Synthetic Interstitial Fluid: 10 mM HEPES, 5 mM KCl, 135 mM NaCl, 1 mM MgCl_2_, 2 mM CaCl_2_, and 10 mM glucose). The final pH for SIF should be 5.0. All components are from Sigma except prostaglandin E2, which is from Cayman Chemicals, Ann Arbor, MI, USA.

### 2.1. Animals, Experimental Groups, and Conditions

Male Sprague–Dawley rats (225–275 g, Harlan Laboratories, Livermore, CA, USA) were used for the experiments. Rats were maintained two per cage in a controlled environment (temperature: 21.5 ± 4.5 °C; relative humidity: 35–55%) under a standard 12 h light/12 h dark lighting cycle. Cage changes occurred twice per week, using standard bedding.

Male C57Bl/6J mice were purchased from Jackson Laboratories (Bar Harbor, MA, USA). They were housed under a standard vivarium condition of 12 h light/dark cycle and ambient temperature with food and water ad libitum. All experiments were started after acclimating the animals for at least 7 days in the facility and were conducted during the light cycle phase (6:30 am to 6:30 pm).

Group assignments were random, and observers were blinded to experimental groups during these studies. Every effort was made to minimize the number of animals and their suffering. Data analysis was carried out by the experimenter blind to the experimental condition of the animals and the identity of treatments. To minimize potential confounder effects, behavior testing was conducted around the same time every day and the order of the experiments was kept consistent in all groups. Initial periosteal injection studies were conducted in rats because their larger size is better suited for detailed anatomical studies of the trigeminal ganglion (TG). Using both species allows for a broader range of experimental techniques, thus facilitating a robust research approach.

#### Ethics Approval

The Veterans Affairs Palo Alto Health Care System (Palo Alto, CA, USA) and Stanford University (Palo Alto, CA, USA) Institutional Animal Care and Use Committee approved all experimental procedures and protocols in accordance with the guidelines of the National Institutes of Health Guide for the Care and Use of Laboratory Animals. The protocol number is CLA1612, approved on 6 January 2022.

### 2.2. Direct Calvarial Periosteal Injection for Behavior Studies

To avoid extracranial tissue damage, the periosteal injection technique was modified from the published study [24]. Briefly, mice were lightly anesthetized (3% isoflurane) and a 27-gauge needle was fitted to a Hamilton syringe which was angled and directly advanced through the skin and underlying galea aponeurotica so that the needle was positioned to deliver the solution into the periosteum overlying the frontal and parietal bones.

Oxytocin or vehicle injection: OT (Sigma) was dissolved in phosphate-buffered saline (PBS; Life Technologies, Carlsbad, CA, USA). Each animal received 50 μL of 20 μg oxytocin (stored at −20 °C until use). In the vehicle control, 50 μL of only PBS was injected.

OTR antagonist administration: L-368,899 is a nonpeptide, desamino-OT analog, and a competitive OTR antagonist [25]. To investigate if a potential pain-attenuating effect of oxytocin could be blocked by a receptor antagonist, animals received a periosteal injection of atosiban, oxytocin receptor antagonist, (0.3 mg diluted in 2 mL saline) or vehicle prior to direct OT (20 μg) injection into calvarial periosteum. A maximum of 50 μL of solution was injected before allodynia testing.

### 2.3. Chemical Activation of Calvarial Periosteum

Based on a previously published study [20] by Zhao et al., an inflammatory mediator (IM) was directly injected into the calvarial periosteum of mice as described above to chemically stimulate the calvarial periosteal afferents. Injection of synthetic interstitial fluid (SIF) was used as a control. Ten mins after injection, animals were deeply anesthetized with isoflurane (5%), and transcardially perfused with ice-cold saline followed by 4% paraformaldehyde. TG was post-fixed in 4% paraformaldehyde for 24 h for immunohistochemical processing.

### 2.4. Retrograde Tracing of Trigeminal Afferents That Innervate the Periosteum (In Vivo) and Systemic Lipopolysaccharide Injection to Establish Inflamed State

Trigeminal afferents innervating the calvarial periosteum were retrogradely labeled using a previously established protocol [20]. Animals (rats) were anesthetized, and the scalp tissue was cut at the midline carefully without damaging the calvarial periosteum. A 2% Fluorogold (FG) solution (in sterile saline) was injected into the periosteum. A small piece of parafilm was then glued onto the periosteum to avoid contact of the tracer with scalp tissue. The skin was sutured, and animals were returned to their home cages for recovery. As we have previously demonstrated that systemic inflammation induces a rapid and robust OTR upregulation [26], some animals received an IP injection of either 1 mL saline or 5 mg/kg lipopolysaccharide (LPS) in 1 mL of saline at 14 days following FG injection. The LPS dose was based on a previously published report established to induce chronic neuroinflammation and pro-inflammatory cytokines in rats [27]. Animals were sacrificed 6 h after saline/LPS administration. The TG and periosteum were harvested and placed in 4% paraformaldehyde for immunohistochemical processing.

### 2.5. Immunohistochemistry

The trigeminal ganglion (TG) and periosteum samples were post-fixed in a 4% paraformaldehyde solution for 24 h. After fixation, the tissues were cryoprotected in 20% sucrose dissolved in phosphate-buffered saline (PBS) for two days at 4 °C, and then rapidly frozen using dry ice. Transverse sections, each 20 µm thick, were then prepared with a cryostat.

Immunostaining procedures used antibodies against phospho-ERK (p-ERK: 1:200, P-p44/42 MAPK, Cell Signaling, Danvers, MA, USA), rabbit anti-oxytocin receptor (OTR; 1:100, ab217212, Abcam), and goat anti-calcitonin gene-related peptide (CGRP; 1:100, ab36001, Abcam). All sections were blocked at room temperature for one hour in PBS containing 8% normal donkey serum and 0.1% Triton X-100, followed by incubation with the primary antibodies for 24 h at 4 °C. Afterward, the tissues were rinsed and incubated with fluorescent secondary antibodies: donkey anti-rabbit Alexa Fluor 488 (1:100, A21206, Invitrogen, CA, USA) to detect p-ERK and OTRs, and donkey anti-goat Alexa Fluor 488 (1:200, A21467, Invitrogen) to detect CGRPs. The sections were then mounted on slides with Antifade Mount containing NucBlue™ Stain (Invitrogen, USA). OTR and CGRP staining was visualized with an epifluorescence microscope (Zeiss, Oberkochen, Germany), using appropriate filter sets. Negative control sections with omitted primary antibodies showed no fluorescence, confirming the specificity of staining. Staining for each group was performed concurrently to allow consistent comparison, and photographs were taken under identical conditions.

To ensure data accuracy, both the imaging and analysis of images were conducted by observers blinded to the experimental conditions. Sections from a control group (secondary antibody only) established the baseline threshold, and any pixel value exceeding this threshold was considered positive staining. The total area of positive staining or the percentage area covered by positive staining in each section was calculated using the NIH’s ImageJ software (https://imagej.net/ij/download.html, accessed on 19 May 2024).

### 2.6. Closed-Head Model of Mild Traumatic Brain Injury (TBI)

The closed-head mild TBI (TBI) and sham procedures were performed as described in previous studies [28,29,30,31]. In summary, a stereotaxic impactor (MyNeurolab, St. Louis, MO, USA) was mounted on a stereotaxic frame (David Kopf Instruments, Tujunga, CA, USA) at a 40-degree angle, using a 5 mm impactor tip. Anesthetized mice (5% isoflurane) were placed in a foam mold and held in a prone position on the stereotaxic frame, with anesthesia maintained at 1.5% throughout the procedure. The stereotaxic arm was positioned to ensure the impact point was consistent and aligned with the S1 somatosensory cortex, just relative to the right eye and ear.

The impact was delivered at a speed of 5.8 to 6.0 m per second, with a dwell time of 0.2 s and an impact depth of 5 mm. Following the procedure, the mice were allowed to recover from anesthesia on a warming pad before being returned to their home cages. No skull fractures were observed, which is consistent with previous studies using similar impact parameters.

For the sham group, the same procedure was followed, except that the impactor was discharged into the air, not contacting the animal. Group sizes for these experiments, ranging from 4 to 6 mice, were calculated to have about 80% statistical power to detect 25% changes at the *p* < 0.05 significance level. Allodynia testing (periorbital) was conducted on day 2 post-injury, while the same mice underwent bright-light stress (BLS) analysis on day 40.

### 2.7. Mechanical Nociceptive Assessment of Periorbital Allodynia

Protocol is adapted from previous studies [32,33]. Mice were individually placed in elevated Plexiglass chambers with mesh flooring and were allowed to acclimate for 2 h daily over a 3-day period. Cephalic (periorbital) allodynia was measured following each 2 h acclimation session. Testing commenced on day 0 (pre-TBI baseline) and was conducted at regular intervals thereafter. A 0.4 g (3.61) von Frey filament was used for testing; it was applied 10 times, with each application held for 5 s and a 20–30-s interval between each application. A positive response was defined as swiping at the face, shaking the head, or turning away from the stimulus. Reactions like running away or rearing up were not counted as positive responses. The frequency of response was calculated as the percentage of positive responses out of the 10 applications, with the formula: ([number of positive responses/10] × 100%). Only animals with an average baseline response frequency of approximately 30% were included in the final experiments.

### 2.8. Bright-Light Stress (BLS) Challenge

To assess BLS, mice were allowed to recover to baseline mechanical thresholds after the TBI. The BLS challenge protocol was based on previously published studies [32,33]. Briefly, BLS was induced by LED strips (1000 lux output). At weeks 4 and 8, animals underwent BLS challenge protocol. Unrestrained mice were exposed for 15 min to BLS that were placed on both sides of their Plexiglass home cages. Similar to previously published study, BLS for 15 min produced mild stress arising from endogenous mechanisms and did not elicit significant CA in naïve or sham mice [32,33].

### 2.9. Statistical Methods

Data are presented as the mean and standard error of the mean (SEM). All statistical analyses were conducted in Graphpad Prism 7 (GraphPad Software, La Jolla, CA, USA). Data for all the immunohistochemical analysis experiments and tests were analyzed using unpaired Student’s *t*-tests. Two-way repeated measures (RM) analysis of variance (ANOVA) was performed for periorbital allodynia, BLS, and immunostaining experiments. Tukey’s or Sidak’s post hoc test for multiple comparisons was used to assess differences between the groups within each time point. Statistical significance was established a priori at 95% (*p* < 0.05).

### 2.10. Data Availability

The datasets generated and/or analyzed during the current study are available from the corresponding author on request.

## 3. Results

### 3.1. Selective Chemical Stimulation of Periosteal Tissue Results in Rapid Activation of TG Neurons

A local application of a mixture of nociceptive IM (inflammatory mediators) containing histamine, serotonin, bradykinin, and prostaglandin E2, has previously been shown to promote activation and sensitization of dural and periosteal nociceptors [20,34]. To test whether local application of the IM could lead to rapid activation of the TG neurons, the IM solution was injected selectively into the periosteum of mice. A synthetic interstitial fluid (SIF) was used as a control. An immunohistochemical (IHC) analysis of phospho-ERK (p-ERK) levels as an indicator of neuronal activation showed significantly increased levels 10 min after injection in the IM group compared to the SIF group, *p* < 0.001, n = 4 per group (Figure 1).

### 3.2. Calvarial Periosteal Trigeminal (CPT) Afferents Express CGRPs and OTRs

In order to provide evidence for the periosteum as a targeted injection site for oxytocin, we examined whether TG afferents innervating the periosteum expressed OTRs and/or CGRPs. In non-inflamed rats, the periosteum (Figure 2a,b) shows clear trigeminal afferents expression of OTRs and/or CGRPs as well as co-expression of OTRs and CGRPs in some afferents. The quantification showed significantly increased OTR (*p* = 0.011) and CGRP (*p* = 0.0007) percentage area of positive staining in the LPS vs. saline group (Figure 2c,d).

### 3.3. Retrogradely Labeled Calvarial Periosteal (CPT) Cell Bodies in TG Express CGRPs and/or OTRs

To evaluate whether the TG neurons innervating the periosteum also expressed OTRs and/or CGRPs, we examined the OTR and CGRP immunoreactivity in TG neurons retrogradely labeled with an FG fluorescent tracer. Numerous FG-labeled neurons that were immunoreactive for OTRs and/or CGRPs were seen to be present in the TG. The TG sections revealed extensive retrograde tracer labeling (in blue) of the calvarial periosteum trigeminal (CPT) neurons, which concurrently expressed oxytocin receptors (OTRs) in green and calcitonin gene-related peptides (CGRPs) in magenta (Figure 3a). The quantification showed significantly increased OTR (*p* < 0.0001) and CGRP (*p* = 0.0025) percentage area of positive staining in the LPS vs. saline group (Figure 3b,c).

### 3.4. Significant Increase in OTR Expression in TG Harvested from mTBI Mice Compared to Sham

To evaluate if an mTBI-induced inflammation would impact OTR expression, we investigated the expression of OTRs in the mTBI vs. sham mTBI groups of mice. The results show a significant increase in OTR levels in the mTBI group compared to the sham (*p* < 0.05, n = 3 per group) (Figure 4).

### 3.5. OT Injection into Periosteum Reduced P/PTH-Related Pain

The periosteum injections of oxytocin significantly reduced allodynia in the model of PTHs as well as persistent allodynia in the model of PPTHs for up to one hour post-injection (Figure 5a,b). On day 2 post-mTBI, the mice received oxytocin injections directly into the periosteum followed by mechanical testing for 5 h after injection. On day 2 and day 40 post-mTBI, mice (n = 4–6 per group) received 50 μL of oxytocin (20 μg) or vehicle injection directly into the periosteum or oxytocin injection into the base of the neck (n = 5) to control for injection location. In addition, on day 40 post-mTBI, the same animals were exposed to 15 min of BLS (bright-light stress) which has been shown to re-establish allodynia in mTBI animals but is ineffective in sham animals [32]. An oxytocin injection into the periosteum reduced both the early and persistent (post-BLS) allodynia for up to one hour post-injection (*p* < 0.0001). By 2 h post-injection, allodynia returned to pre-injection but post-mTBI/BLS levels. As an additional control for systemic (non-periosteal) effects, the injection of oxytocin into the neck muscle did not induce any analgesia at either time point, demonstrating that the analgesic effect of oxytocin injection was specific to periosteal innervation.

### 3.6. OTR Antagonism Blocks Analgesic Effect of Periosteal Oxytocin Injection

In a separate set of cohorts, the same protocol described above was followed, but the OTR receptor antagonist L-368,899 hydrochloride is a nonpeptide, desamino-OT analog, and a competitive OTR antagonist (50 μL of 50 μg L-368,899 diluted in saline) was applied immediately before oxytocin application into the periosteum on day 2 and day 40 post-mTBI (n = 4–6 per group). A pre-injection of an antagonist vehicle followed by 20 μg of oxytocin into the periosteum reduced both early and persistent (post-BLS) allodynia, whereas the pre-injection with an OTR antagonist (L-368,899), followed by oxytocin completely blocked the anti-allodynic effect of oxytocin (*p* < 0.0001) (Figure 6a,b). As an additional control, the injection of an OTR antagonist (L-368,899) following an antagonist vehicle injection did not have any effect on either early or persistent allodynia.

## 4. Discussion

Post-traumatic headaches (PTHs) are difficult to treat mainly due to a lack of understanding of the pathophysiology of mTBI sequala leading to headaches. Although previous studies have suggested that the periosteum plays a role in PTHs and migraines, the effect of drug delivery into the periosteum such as oxytocin into the periosteum has not been previously explored. Therefore, this study investigated whether the periosteum can be used as a therapeutic site to deliver drugs such as oxytocin. Our main observations were that: (1) the cell bodies of CP afferents reside in the TG and express both OTRs and CGRPs, (2) the inflammatory chemical stimulation of the periosteum leads to rapid activation of the TG neurons (phospho-ERK (p-ERK) expression), (3) mTBI-induced inflammation increased the OTR expression compared to the sham group, and (4) the administration of oxytocin into the periosteum on day 2 and day 40 blocked cutaneous allodynia for up to one hour post-administration for both acute and persistence phases in the PTH model—an effect that was preventable by the administration of an OTR antagonist. Taken together, our observations suggest that periosteal trigeminal afferents contribute to post-TBI craniofacial pain, and that periosteum tissue can be used as a potential local target for therapeutics such as oxytocin.

In a prior investigation by Zhao et al. [20], it was shown that inflammatory stimulation of periosteal afferents led to periorbital tactile hypersensitivity, indicating a potential involvement of periosteum activation and sensitization in triggering headaches. Our study corroborates previous findings, demonstrating that the inflammatory stimulation of the periosteal afferents rapidly activated the TG neurons. To identify neuronal activation following the inflammatory (noxious) stimulation, we utilized phosphorylated extracellular signal-regulated kinase (pERK) labeling as a marker, given its established induction within minutes of noxious stimulation [35].

Multiple studies have previously suggested that periosteum may play a role in headaches [10,12,20]. For example, Schueler et al. and Zhao et al., demonstrated using neural tracing that a proportion of the meningeal afferents innervate extracranial tissue and these afferents have slowly conducting axons, with nociceptive function [12,20]. The calvarial periosteum is reported to be innervated by small- and medium-sized neurons in the TG [20] with a network of sensory (peripherin-positive) and pain (CGRP- and TRPV1-positive) fibers [10]. Furthermore, a recent study by Gferer et al. [19] demonstrates that onabotulinumtoxinA alters the expression of inflammatory genes largely in the periosteum, minimally in the muscle, and not at all in the fascia in chronic headache patients. The study suggests that in some patients with chronic migraines, inflammation in the periosteum contributes to the pathophysiology of headaches and that the therapeutic effect of onabotulinumtoxinA could be caused by reducing inflammation in the periosteum [19]. Here, we provide further evidence that the periosteum may play a role in headaches, by demonstrating that the periosteal afferents express CGRP-positive fibers that co-label with OTRs. The expression of CGRP-positive and OTR-positive fibers was significantly increased after inflammation, suggesting that the periosteal afferents may play an important role in the extracranial impacts of headache generation and therapy. Although previous studies suggest that CGRPs play a role in migraine pathophysiology and post-traumatic headaches (PTHs), the link between CGRPs and periosteal nociceptive afferents should be interpreted with caution. While CGRPs are known to mediate pain and inflammation, they also have other roles in vascular regulation and neurogenic inflammation. This broader functional context means that CGRP’s involvement in migraines and PTHs may be part of a more complex network of factors, not solely due to nociceptor sensitization. Additionally, while CGRPs have been implicated in the development of migraines and PTHs, other neuropeptides and neurotransmitters also contribute to these conditions. Therefore, focusing solely on CGRPs might overlook other critical mechanisms that influence pain pathways.

It has been previously demonstrated that painful inflammation including facial electro-cutaneous stimulation and adjuvant-induced inflammation of the jaw joints dramatically enhanced OTR expression on the TG neurons [26,36]. In this study, we demonstrate that inflammation due to mTBI led to a persistent increase in OTRs in the TG, corroborating previous findings [26,36]. The current results demonstrate that the level of OTR expression in the trigeminal system, specifically in the TG and periosteum, is dynamic with systemic inflammation and an mTBI increases the expression of the OTR protein. Although the intricate regulation of OTR transcription remains not fully elucidated, it is noteworthy that three response elements on the OTR gene promoter can bind to interleukin-6, likely contributing to the upregulation observed during inflammation, at least within the central nervous system (CNS) [37].

Our study illustrates the potential of delivering oxytocin directly into the periosteum to alleviate mTBI-induced allodynia during both the immediate and prolonged phases. Furthermore, when applied in vitro to trigeminal ganglion (TG) neurons or to the dura, oxytocin inhibits the firing of these neurons and the release of calcitonin gene-related peptides (CGRPs) [26]. Earlier investigations involving rodents with mild traumatic brain injuries (mTBIs) have documented a mechanism dependent on the calcitonin gene-related peptides (CGRPs), which triggers a sensitized state likely responsible for both immediate and prolonged post-traumatic headaches (PTHs) [32]. In addition, continuous sequestration of the CGRP has been reported to prevent mTBI-related acute PTHs as well as stress-induced persistent PTHs [16]. The oxytocin mitigated both reactive and spontaneous, ongoing non-reactive pain following a mild traumatic brain injury (TBI) for a duration of at least 3–4 h subsequent to the intranasal application [23]. The analgesic impact of oxytocin is likely attributable to a reduction in trigeminal ganglion (TG) neuron excitability. This decline in neuronal excitability appears to be facilitated, at least partially, by an augmentation in the density of voltage-gated potassium (Kv) channels [21]. Although the current study does not provide conclusive evidence, the presence of OTRs on CGRP immunoreactive TG neurons and periosteal afferent fibers provides initial support that the inhibition of nociceptive afferent neurons may contribute to the analgesic effects observed. Here, we show that the application of oxytocin inhibits both the PTHs and persistent PTH-induced allodynia. Furthermore, we confirmed that the analgesic effect was due to oxytocin by using a selective OTR antagonist. The co-administration of an OTR antagonist abolished the analgesic effect of oxytocin on mTBI-induced allodynia and BLS-induced allodynia. Future studies to investigate the specific intracellular signaling mechanism through which oxytocin injected into the calvarial periosteum mediates its analgesic effect to inhibit P/PTH-related pain are warranted.

We acknowledge that a limitation of our study is the exclusive use of male animals, considering that sex is a recognized biological variable, and females are known to exhibit a heightened risk for post-traumatic headaches (PTHs) following a mild traumatic brain injury (mTBI). Consequently, future research exploring sex-dependent responses is justified.

## 5. Conclusions

The findings outlined in this study add to our understanding of the underlying mechanism of acute PTHs and persistent PTHs and the role of calvarial periosteum in these processes. The data suggest that calvarial periosteal trigeminal afferents contribute to post-TBI craniofacial pain and that periosteum tissue can be used as a potential local target for therapeutics such as oxytocin.

## Figures and Tables

**Figure 1 pharmaceutics-16-00760-f001:**
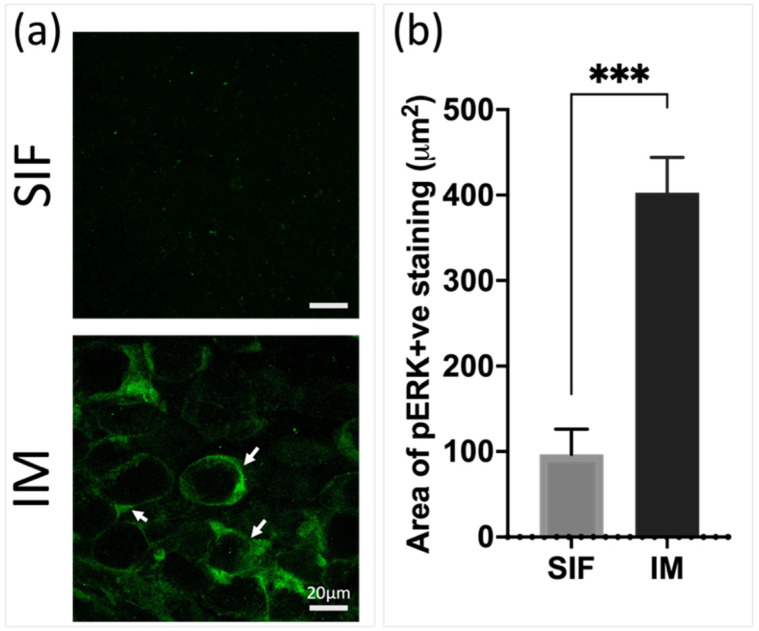
Chemical stimulation using inflammatory mediators (IM) injection into periosteum rapidly and significantly activated cells, p-ERK expression, in the TG compared to control (SIF). (**a**) Representative IHC images of SIF vs IM, (**b**) Quantification of area of pERK positive staining, scale bar = 20 μm. n = 4, *t*-test, *** *p* < 0.001. Arrows show positive pERK staining.

**Figure 2 pharmaceutics-16-00760-f002:**
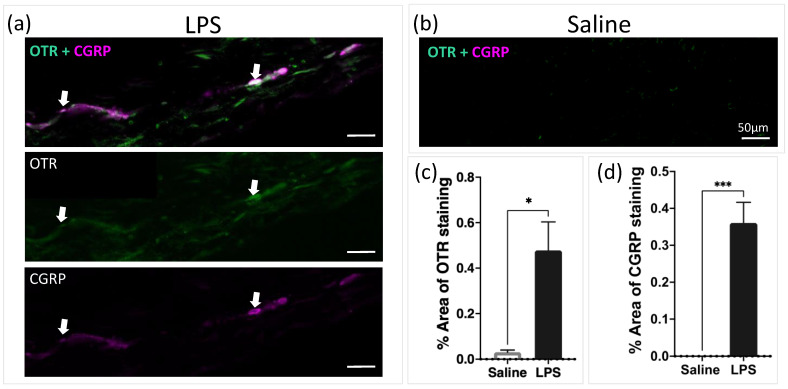
(**a**,**b**) Representative image showing expression of OTRs and CGRPs in calvarial periosteum of (**a**) LPS-injected and (**b**) saline injected animals. Scale bar = 50 μm. (**c**,**d**) Quantification of percentage area of positive staining of OTRs (* *p* = 0.011) and CGRPs (*** *p* = 0.0007) in periosteum harvested from control and inflamed animals show significant increase in inflamed group (n = 4 per group). Arrows show positive staining.

**Figure 3 pharmaceutics-16-00760-f003:**
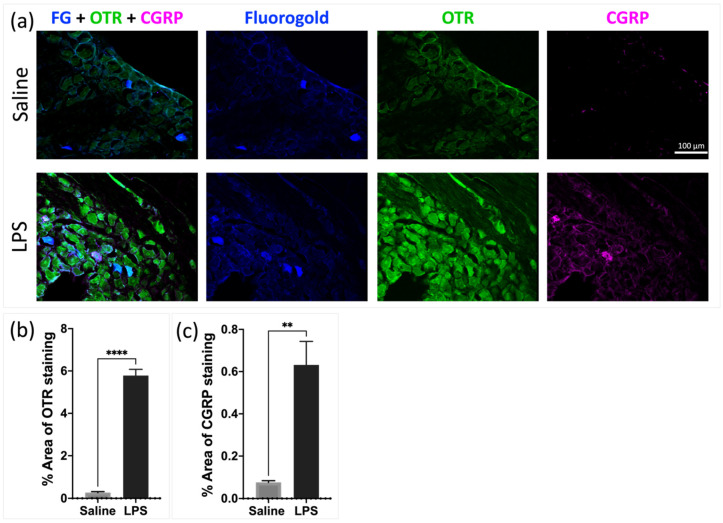
Fluorogold tracer injected into calvarial periosteum shows staining in the TG, with co-labeling of fluorogold, OTRs, and CGRPs. Scale bar = 100 μm for (**a**), insert 25 μm. (**b**,**c**) Quantification of percentage area of positive staining of OTRs (**** *p* < 0.0001) and CGRPs (** *p* = 0.0025) in TG harvested from control and inflamed animals show significant increase in inflamed group (n = 4 per group).

**Figure 4 pharmaceutics-16-00760-f004:**
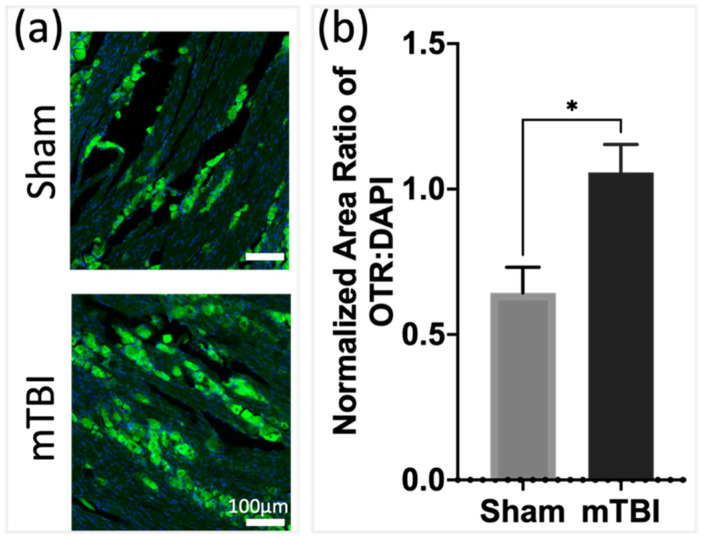
Mild TBI (17 days after) produces significant increase in OTR immunofluorescence in TG (**a**). Scale bar = 100 μm. (**b**) Quantification of area of OTR/DAPI shows significant increase in mTBI group compared to sham, *t*-test, * *p* < 0.05 (n = 3).

**Figure 5 pharmaceutics-16-00760-f005:**
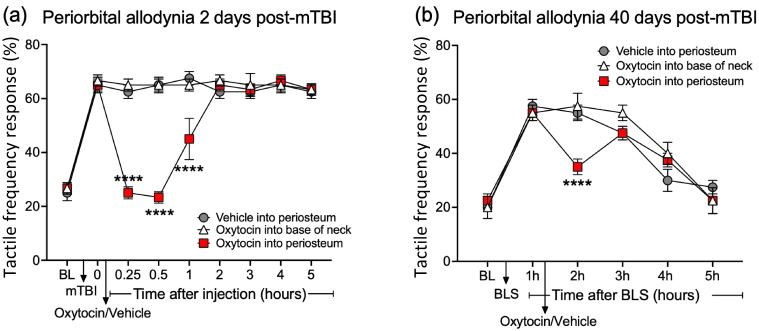
Early and persistent mTBI-related allodynia and effect of direct oxytocin (OT) or vehicle injection; Cranial withdrawal frequency in response to von Frey monofilament stimulation was collected followed by induction of mTBI. On (**a**) day 2 and (**b**) day 40 post-mTBI (and BLS exposure), mice received 50 μL of oxytocin (20 μg) or saline vehicle injection into periosteum or OT injection into base of neck (n = 4–6 per group). OT injection into periosteum significantly reduced allodynia for up to one hour post-injection (**** = *p* < 0.0001, Two-way ANOVA, Tukey’s post hoc test); OT injection into neck muscle was ineffective. Graphs show mean S.E.M.

**Figure 6 pharmaceutics-16-00760-f006:**
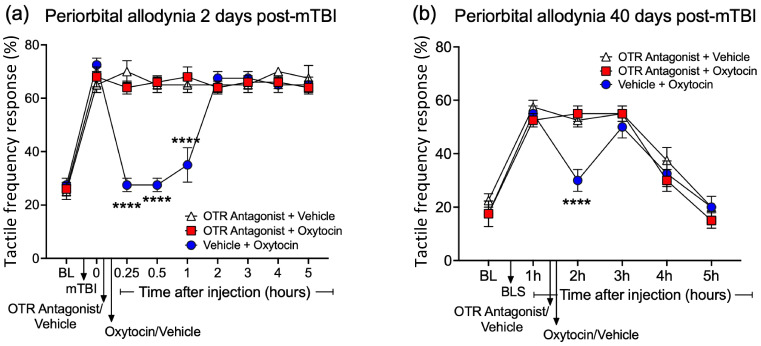
Early and persistent mTBI-related allodynia—effect of OTR antagonist (L-368,899) or vehicle injection into periosteum prior to OT or vehicle injection into periosteum; Frequency of response to cranial von Frey monofilament application was collected. On (**a**) day 2 and (**b**) day 40 post-mTBI (and BLS exposure), mice received OTR antagonist (or vehicle) into periosteum 5 min prior to OT (50 μL of 20 μg OT) or vehicle injection into the periosteum (n = 4–6 per group). Only OT injection into periosteum blocked PTH-related pain for up to one hour post-injection. OTR antagonist injection into periosteum blocked OT-induced analgesia. Two-way ANOVA (Tukey’s post hoc test). Graphs show mean S.E.M. (**** *p* < 0.0001).

## Data Availability

Data will be made available.

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
