# Peer review of "Oxytocin Receptors on Calvarial Periosteal Innervation: Therapeutic Target for Post-Traumatic Headache?"

_pharmaceutics, 2024, doi:10.3390/pharmaceutics16060760_

Round 1

Reviewer 1 Report

Comments and Suggestions for Authors
  • The authors need to provide more details regarding the primary research objectives at the end of the introduction.
  • A concise summary of the specific aims of the study to enhance the understanding of the subsequent sections.
  •  
  • The authors briefly mention the potential therapeutic effect of oxytocin on periosteal afferents, but further explain the underlying mechanisms of the rationale for this hypothesis.
  • Exploring the known mechanisms by which oxytocin modulates neuronal excitability and pain perception, especially in the context of trigeminal nerve function, would provide additional insight into the proposed therapeutic strategy.
  • Previous studies linking calcitonin gene-related peptide (CGRP) and periosteal nociceptive afferents to migraine and PTH. The authors must acknowledge any potential limitations or alternative interpretations of these findings. Providing a balanced discussion of the existing literature will enhance the credibility of the rationale for the current study and help contextualize its significance within the broader research landscape

Author Response

The authors need to provide more details regarding the primary research objectives at the end of the introduction.

We thank the reviewer for drawing attention to adding more details regarding the primary research objectives at the end of the introduction. As per the recommendation, we have now inserted the objectives.

Introduction:

The objective of this study is to determine whether calvarial periosteal neurons express OTR and if they are a useful target for inhibition of PTH related pain.  

A concise summary of the specific aims of the study to enhance the understanding of the subsequent sections.

As per the recommendation, we have now edited a concise summary of specific aims.

The study's specific aims are to determine whether: (1) periosteal stimulation activates trigeminal ganglion (TG) neurons, (2) trigeminal neurons and fibers in the periosteum express oxytocin receptor (OTR) and/or calcitonin gene-related peptide (CGRP), (3) inflammation from mild traumatic brain injury (mTBI) increases OTR expression, and (4) post-traumatic headache (PTH)-related pain can be reduced by injecting oxytocin into periosteal tissue after mTBI, with OTR antagonists negating this analgesic effect.

The authors briefly mention the potential therapeutic effect of oxytocin on periosteal afferents, but further explain the underlying mechanisms of the rationale for this hypothesis.

The rationale for this hypothesis lies in the known effects of oxytocin on nociception and inflammation. Oxytocin has been shown to modulate pain by interacting with OTRs on sensory neurons, leading to a reduction in pain signaling. Additionally, oxytocin has anti-inflammatory properties, which could help to lower the inflammatory response following mTBI, a common contributor to craniofacial pain. By reducing primary afferent activity and sensitization, oxytocin delivery to the calvarial periosteum might provide a targeted therapeutic approach for managing craniofacial pain after mTBI. This mechanism could help to minimize the intensity of post-traumatic headaches and related discomfort, offering a promising intervention for those suffering from these conditions.

Exploring the known mechanisms by which oxytocin modulates neuronal excitability and pain perception, especially in the context of trigeminal nerve function, would provide additional insight into the proposed therapeutic strategy.

We have added sentences providing additional detail as to the means by which oxytocin may inhibit the firing of trigeminal nociceptive neurons. 

In particular, we have found that oxytocin increased the resting membrane potential and rheobase of trigeminal neurons taken from rats with a pre-existing craniofacial injury, both of which indicate a clear decrease in excitability21, likely mediated by an increase in potassium current22.

Previous studies linking calcitonin gene-related peptide (CGRP) and periosteal nociceptive afferents to migraine and PTH. The authors must acknowledge any potential limitations or alternative interpretations of these findings. Providing a balanced discussion of the existing literature will enhance the credibility of the rationale for the current study and help contextualize its significance within the broader research landscape

Although previous studies suggest that CGRP plays a role in migraine pathophysiology and post-traumatic headache (PTH), the link between CGRP and periosteal nociceptive afferents should be interpreted with caution. While CGRP is known to mediate pain and inflammation, it also has other roles in vascular regulation and neurogenic inflammation. This broader functional context means that CGRP's involvement in migraine and PTH may be part of a more complex network of factors, not solely due to nociceptor sensitization. Additionally, while CGRP has been implicated in the development of migraine and PTH, other neuropeptides and neurotransmitters also contribute to these conditions. Therefore, focusing solely on CGRP might overlook other critical mechanisms that influence pain pathways.

Reviewer 2 Report

Comments and Suggestions for Authors

This study, though not original, provides incremental knowledge about oxytocin receptor (OTR as a target to treat headaches after mild TBI (mTBI). However, the significance of this supporting study is high because it can lead to the treatment of headache after TBI. The manuscript is strengthened by using both OT and OTR antagonist in an appropriate mTBI animal model. Overall, the research is well-designed, and the manuscript is well-written and presented. All figures are appropriate. Discussion is relevant and appropriate. Indicating that OT may be reducing headache by inhibiting TG neurons excitability seems to be interesting and logical. I would like to extend some minor comments.

1.       Add some justification for the use of both rats and mice.

2.       Include an appropriate reference for the preparation of inflammatory mediators (IM) in the materials and methods section.

3.       It is good that the authors acknowledge the limitations of using only male animals. However, I recommend including studies on female animals (rats or mice) also. This addition will help to take the study levels up and develop OT-based headache therapy for TBI.

Author Response

We thank the reviewers for the thorough review and important questions/comments that helped refine the revised manuscript. We appreciate that the reviewers noted the original manuscript is well-designed, and rigorous.

Point by point responses are listed below (response in blue font) and the manuscript text edits are italicized here, and changes tracked in the manuscript (.doc file).

Reviewer 1

The authors need to provide more details regarding the primary research objectives at the end of the introduction.

We thank the reviewer for drawing attention to adding more details regarding the primary research objectives at the end of the introduction. As per the recommendation, we have now inserted the objectives.

Introduction:

The objective of this study is to determine whether calvarial periosteal neurons express OTR and if they are a useful target for inhibition of PTH related pain.  

A concise summary of the specific aims of the study to enhance the understanding of the subsequent sections.

As per the recommendation, we have now edited a concise summary of specific aims.

The study's specific aims are to determine whether: (1) periosteal stimulation activates trigeminal ganglion (TG) neurons, (2) trigeminal neurons and fibers in the periosteum express oxytocin receptor (OTR) and/or calcitonin gene-related peptide (CGRP), (3) inflammation from mild traumatic brain injury (mTBI) increases OTR expression, and (4) post-traumatic headache (PTH)-related pain can be reduced by injecting oxytocin into periosteal tissue after mTBI, with OTR antagonists negating this analgesic effect.

The authors briefly mention the potential therapeutic effect of oxytocin on periosteal afferents, but further explain the underlying mechanisms of the rationale for this hypothesis.

The rationale for this hypothesis lies in the known effects of oxytocin on nociception and inflammation. Oxytocin has been shown to modulate pain by interacting with OTRs on sensory neurons, leading to a reduction in pain signaling. Additionally, oxytocin has anti-inflammatory properties, which could help to lower the inflammatory response following mTBI, a common contributor to craniofacial pain. By reducing primary afferent activity and sensitization, oxytocin delivery to the calvarial periosteum might provide a targeted therapeutic approach for managing craniofacial pain after mTBI. This mechanism could help to minimize the intensity of post-traumatic headaches and related discomfort, offering a promising intervention for those suffering from these conditions.

Exploring the known mechanisms by which oxytocin modulates neuronal excitability and pain perception, especially in the context of trigeminal nerve function, would provide additional insight into the proposed therapeutic strategy.

We have added sentences providing additional detail as to the means by which oxytocin may inhibit the firing of trigeminal nociceptive neurons. 

In particular, we have found that oxytocin increased the resting membrane potential and rheobase of trigeminal neurons taken from rats with a pre-existing craniofacial injury, both of which indicate a clear decrease in excitability21, likely mediated by an increase in potassium current22.

Previous studies linking calcitonin gene-related peptide (CGRP) and periosteal nociceptive afferents to migraine and PTH. The authors must acknowledge any potential limitations or alternative interpretations of these findings. Providing a balanced discussion of the existing literature will enhance the credibility of the rationale for the current study and help contextualize its significance within the broader research landscape

Although, previous studies suggest that CGRP plays a role in migraine pathophysiology and post-traumatic headache (PTH), the link between CGRP and periosteal nociceptive afferents should be interpreted with caution. While CGRP is known to mediate pain and inflammation, it also has other roles in vascular regulation and neurogenic inflammation. This broader functional context means that CGRP's involvement in migraine and PTH may be part of a more complex network of factors, not solely due to nociceptor sensitization. Additionally, while CGRP has been implicated in the development of migraine and PTH, other neuropeptides and neurotransmitters also contribute to these conditions. Therefore, focusing solely on CGRP might overlook other critical mechanisms that influence pain pathways.

Reviewer 2

This study, though not original, provides incremental knowledge about oxytocin receptor (OTR as a target to treat headaches after mild TBI (mTBI). However, the significance of this supporting study is high because it can lead to the treatment of headache after TBI. The manuscript is strengthened by using both OT and OTR antagonist in an appropriate mTBI animal model. Overall, the research is well-designed, and the manuscript is well-written and presented. All figures are appropriate. Discussion is relevant and appropriate. Indicating that OT may be reducing headache by inhibiting TG neurons excitability seems to be interesting and logical. I would like to extend some minor comments.

We thank the reviewer for their positive and insightful comments.

Add some justification for the use of both rats and mice.

We thank the reviewer for the suggestion. We have edited accordingly.

Materials and method:

Initial periosteal injections studies were conducted in rats for their larger size that are better for detailed TG anatomical studies. Using both species allowed for a broader range of experimental techniques, thus facilitating more robust and multidimensional research.

Include an appropriate reference for the preparation of inflammatory mediators (IM) in the materials and methods section.

The reference is now included.  Based on previously published study20 by Zhao et. al.,

It is good that the authors acknowledge the limitations of using only male animals. However, I recommend including studies on female animals (rats or mice) also. This addition will help to take the study levels up and develop OT-based headache therapy for TBI.

We thank the reviewer for their insightful comments and for highlighting the importance of considering both male and female animals in preclinical studies. We agree that including female subjects could enhance the generalizability of our findings and contribute to a broader understanding of the potential effects of OT-based therapies for traumatic brain injury (TBI)-related headaches.

Unfortunately, due to specific constraints in our current research design, such as time, and budget, we are unable to include female animals in this study. Our initial focus on male animals was driven by the need to establish a consistent baseline and control for variables such as hormonal fluctuations that could affect our outcomes. Given these factors, we opted for a more controlled approach to ensure the robustness of our data.

We acknowledge that this limitation may affect the scope of our conclusions and plan to address this in future studies, where we hope to explore potential gender-based differences in TBI-related headaches and OT-based treatments. We appreciate your suggestion and will keep it in mind as we continue to refine our research approach.

Round 2

Reviewer 1 Report

Comments and Suggestions for Authors

Well done